# Fer and FerT: A New Regulatory Link between Sperm and Cancer Cells

**DOI:** 10.3390/ijms24065256

**Published:** 2023-03-09

**Authors:** Uri Nir, Elina Grinshtain, Haim Breitbart

**Affiliations:** The Mina & Everard Faculty of Life Sciences, Bar-Ilan University, Ramat-Gan 5290002, Israel; uri.nir@biu.ac.il (U.N.); elina.grinshtain@biu.ac.il (E.G.)

**Keywords:** Fer, FerT, sperm, cancer cells, signaling

## Abstract

Fer and its sperm and cancer specific variant, FerT, are non-receptor tyrosine kinases which play roles in cancer progression and metastasis. Recent studies have shed light on the regulatory role of these kinases in ensuring proper sperm function. Comparison of the regulatory cascades in which Fer and FerT are engaged in sperm and cancer cells presents an interesting picture, in which similar regulatory interactions of these enzymes are integrated in a similar or different regulatory context in the two cell types. These diverse compositions extend from the involvement of Fer in modulation of actin cytoskeleton integrity and function, to the unique regulatory interactions of Fer with PARP-1 and the PP1 phosphatase. Furthermore, recent findings link the metabolic regulatory roles of Fer and FerT in sperm and cancer cells. In the current review, we discuss the above detailed aspects, which portray Fer and FerT as new regulatory links between sperm and malignant cells. This perspective view can endow us with new analytical and research tools that will deepen our understanding of the regulatory trajectories and networks that govern these two multi-layered systems.

## 1. Introduction

Fer and FerT are non-receptor tyrosine kinases (NRTK), which belong to the unique feline sarcoma kinase (Fes) and feline sarcoma-related (Fer) kinases family of NRK [1]. Members of this group are characterized by a kinase domain (KD) residing in the C-terminal region, which is preceded by an SH2 domain and a long N-terminal tail bearing three coiled-coil domains [1,2]. The N-terminal region also contains an FCH (Fps/Fes/Fer/CIP4 Homology) motif which, together with the first coiled-coil elements, forms an F-BAR domain. This F-Bar domain is similar to other BAR domains, which are modules that function in the regulation of actin cytoskeleton and signaling pathways [3]. The Fes/Fer family members were shown to be engaged in the transmission of outside–inside, receptor mediated signaling cascades [1,4]. However, the dispersion of Fer throughout the cell cytoplasm, including the mitochondria [5], suggests its involvement in divers signaling pathways. Specifically, the Fer kinase was shown to promote cell proliferation [6], invasion, and migration [7]. Concomitantly, Fer was found to be involved in the development and progression of malignant processes, and in the onset of metastasis [8,9]. Accordingly, high expression of Fer has been shown to serve as an independent prognostic factor that correlates with worse overall survival of triple-negative breast cancer (TNBC) [6], non-small cell lung cancer (NSCLC), and pancreatic ductal adenocarcinoma cancer (PDAC) [10]. A recombinant protein produced by the fusion between the first 13 exons from the mannosidase α class 2A member 1 gene (MAN2A1) and the last 6 exons of the *Fer* tyrosine kinase gene is termed MAN2A1-FER. This fusion protein retains the tyrosine kinase activity of Fer and was detected in liver tumors, esophageal adenocarcinoma, glioblastoma multiforme, prostate and non-small cell lung cancers, and ovarian tumors, but not in normal nontumor tissues [11]. Furthermore, the MAN2A1-FER promotes the proliferation, invasiveness, and metastatic dissemination of the expressing malignant cells [11]. A fusion transcript between the IL-2 inducible T-cell kinase-RNA (*itk*), and the Fer RNA (*fer*), termed-ITK-FER, was detected in a follicular helper T-cell-peripheral, T-cell lymphoma [12,13]. The pro-oncogenic activity of Fer was shown to be linked to various regulatory pathways, some of which seem to differ among distinct cancer types. Fer activates the Wnt/β-catenin signaling pathway to propel the growth and metastatic dissemination of melanoma tumors [6]. Upregulation of the anti-apoptotic protein Bcl-2, and the mitogen-activated protein kinase-P38 by Fer, were linked to the increased survival and proliferation of bladder cancer cells [14]. Fer promotes also the deregulated proliferation of ovarian cancer cells by phosphorylating the insulin receptor substrate four (IRS4), thereby enabling it to recruit the PIK3R2/p85β-subunit of PI3K, and to activate the PI3K-AKT pathway [8]. The Fer driven metastatic dissemination of ovarian cancer cells depends on the Fer mediated phosphorylation and activation of the Hepatocytes Growth Factor Receptor (HGFR/MET) in a ligand-independent manner [7]. This leads to the activation of the RAC1-PAK1 signaling pathway, which is required for the metastatic potential of ovarian cancer cells [15,16]. In pancreatic ductal adenocarcinoma, Fer potentiated the migration and invasion of the malignant cells by activating the STAT3/MMP2 cascade [17]. Fer dependent stimulation of cell migration was also found to be mediated by the Fer driven activation of JNK [17]. Interestingly, the ability of Fer to regulate endosomal recycling was also found to serve as a mechanism that underlies the Fer-dependent breast cancer invasion [17]. At the molecular level, Fer was found to enhance the recycling of the endosomal EGF receptor (EGFR) to the cell surface, by phosphorylating PKCδ which then attenuates the delivery of EGFR to the lysosome for degradation. The consequent upregulated EGFR signaling was found in tumors of 25% TNBC patients and is presumed to contribute to the aggressiveness of the disease [17]. Finally, a recent work revealed a new mechanism through which Fer propels the proliferation, migration, and invasion of breast cancer cells. Molecularly, Fer phosphorylates the transcriptional corepressor one (SKOR1) on tyrosine-234. This phosphorylation enables SKOR1 to bind Smad3 and to promote the Smad2/Smad3 signaling pathway. It turns out that SKOR1 is a key mediator of the Fer-dependent progression of high-risk breast cancers [18]. Hence, Fer is an important oncogenic factor which is engaged in multiple signaling cascades and can promote malignant tumor progression through diverse regulatory pathways. Of note is the truncated variant of Fer–FerT [19], which bears the KD and the SH2 domains of Fer, linked to a 44 amino-acid (aa) long N-terminal tail [2,19]. Thus, FerT lacks the FCH motif and its following coiled-coil domains, which compose the F-BAR domain of Fer. Fer and FerT may therefore differ in their ability to regulate actin cytoskeleton functioning. Unlike Fer, the expression of FerT is restricted to meiotic and post-meiotic spermatogenic cells [5,20]. This meiotic and post-meiotic expression profile of FerT is directed and driven by an intronic promoter which resides in intron 10 of the *fer* locus and is activated by the spermatogenic specific, Boris transcription factor [21]. An intriguing finding was our observation that, although not being found in normal somatic cells, the spermatogenic specific FerT is expressed in various cancer cell types, always together with Fer. Furthermore, the expression of FerT is increased in colon cancer metastatic lesions [5,21]. Thus, FerT is a sperm and cancer specific tyrosine kinase which is co-expressed with Fer in both cell types. This link between sperm and cancer puzzled us and motivated us to further explore the roles of Fer and FerT in sperm and cancer cells. Several studies suggested the involvement of Fer and FerT in modulating F-actin microtubules networks [22] in spermatogenic cells. In the current review, we describe and discuss regulatory cascades in which Fer and FerT are engaged in sperm and cancer cells, and the common aspects of these roles in sperm and malignant cells.

## 2. Fer and FerT: Newly Established Links between Sperm and Cancer Cells

A breakthrough in resolving part of the enigma referring to the common role of Fer and FerT in sperm and malignant cells came when we found out that knockdown of Fer or FerT led to an increase in ROS levels in malignant cells [23]. One of the cellular ROS production sites is the mitochondrial electron transport chain (ETC), primarily through complex I (Comp. I) and complex III (Comp. III) [24]. We therefore examined the association of Fer and FerT with the mitochondrial ETC in both spermatogenic and malignant cells. This revealed the association of Fer/FerT with Comp. I in the mitochondria of both spermatogenic and malignant cells. Notably, Fer does not associate with the mitochondrial ETC in normal somatic cells [5]. Furthermore, knockdown of FerT downregulated the Comp. I activity in the mitochondria of malignant cells [5]. Importantly, targeting both Fer and FerT with a selective synthetic inhibitor (E260) directed toward the common KD of these kinase in malignant cells led to mitochondrial depolarization, deformation, and disfunction, followed by a significant decrease in cellular ATP production [25]. The mitochondrial deformation caused by E260 suggests the involvement of Fer/FerT in maintaining key mitochondrial integrity sustaining processes in malignant cells. In compliance with these observations, treatment of sperm cells with the selective Fer/FerT inhibitor, E260, significantly impaired their motility, an effect that could be partially reversed by exposing the treated sperm cells to a high glucose concentration [26]. This suggests that E260 inhibits sperm motility through inhibition of the mitochondrial oxidative phosphorylation (Oxphos.) energy generation process, an effect that can be partially relieved by upregulating the cells’ glycolysis, which can be disengaged from the mitochondrial Oxphos. in sperm cells [27]. The presence and function of Fer/FerT in the mitochondria of sperm and malignant cells guided us to further unravel the common metabolic modules of sperm and cancer cells. To directly corroborate the metabolic role of FerT in the development and progression of oncogenic processes, the expression of the testis specific kinase was enforced in the mitochondria of non-malignant cells. This ectopic expression of FerT endowed the transfected cells with an ability to form aggressively growing, highly vascularized tumors in immunocompromised mice [5]. Thus, the mitochondrial function of FerT is directly linked to the development and progression of malignant processes. Cancer cells, especially at their metastatic state, require high metabolic plasticity, adaptability, and metabolic stress resistance [28]. Moreover, the metabolic disengagement between aerobic glycolysis, which is shunted toward lactic acid production, and mitochondrial OxPhos., in cancer cells necessitates the reprogramming of the metabolic mitochondrial system in malignant cells. For example, the mitochondrial glutamine uptake machinery, which propels the tricarboxylic acids (TCA) cycle independently of glycolysis and its end product, pyruvate, is upregulated in cancer cells [29]. The disengagement between glycolysis and mitochondrial OxPhos endows cancer cells with metabolic plasticity and flexibility through which the elative activation state of the glycolytic pathway and the mitochondrial OxPhos can be modulated and tuned according to the varying metabolic and environmental needs. It is assumed that during metastatic dissemination, cancer cells encounter stressful metabolic challenges that force them to exploit their reprogrammed metabolic plasticity. One major cause of this phenomenon may be the detachment of metastasizing cancer cells from the primary tumor and its micro-environment, thereby becoming disconnected from the blood and glucose supply, leading to the inability of the cells to carry out aerobic glycolysis. Consequently, this process enforces the adoption of an alternative metabolic pathway, most commonly executed in the reprogrammed mitochondrial metabolic system [30]. Sperm cells seem to encounter similar metabolic challenges when moving in the female reproductive tract, along the uterus toward the female fallopian tube, a path in which they are devoid of direct blood supply and nutrients, such as glucose. Under these stressful circumstances, Fer and FerT may potentiate the mitochondrial activity through upregulation of Comp. I, thereby endowing sperm and metastatic cancer cells with an essential, acquired metabolic plasticity, and enabling their survival and function under limited glucose availability. Accordingly, knocking out the *fer/ferT* genes in metastatic, non-small cell lung cancer (NSCLC) cells severely jeopardized their metabolic plasticity and significantly impaired their ability to grow in culture and form tumors in vivo under glucose restrictive conditions [31]. Thus, Fer/FerT endow both sperm and cancer cells with a required metabolic plasticity. As we noted above, inhibition of the Fer/FerT activity with the selective inhibitor, E260, downregulated the mitochondrial activity in both sperm and malignant cells. While this caused motility arrest in sperm [26], in cancer cells it evoked autophagy driven necrotic death [25]. This resulted from simultaneous alteration of parallel pathways following the inhibition of Fer/FerT in malignant cells. For example, impairment of the mitochondrial function led to reduced ATP levels in E260 treated cancer cells and to the consequent activation of the 5′ AMP-activated protein kinase (AMPK) [25,32], which by itself leads to the inhibition of the key metabolic modulator, mTOR [25,33]. Exacerbated depletion of cellular ATP levels was also achieved due to another unexpected regulatory involvement of Fer. Through the application of interactomic and proteomic analyses, we found out that Fer associates with the poly [ADP-ribose] polymerase 1 (PARP-1) enzyme [25] in the nucleus of cancer cells and restrains its activity. This regulatory association between Fer and PARP-1 is mediated through the kinase domain which is common to Fer and FerT [25]. Poly-ADP-ribosylation (PARylation), covalently linking ADP-ribosyl moieties (PARs) to a substrate, is a post-translational modification of proteins, catalyzed by ADP-ribosyl transferases (ART), including the nuclearPARP-1. Originally, studies of PARylation and PARPs focused on DNA damage responses in cancer, but more recent studies revealed diverse roles in a broader array of biological processes [34]. Importantly, ongoing PARylation leads to severe depletion of ATP which is one of the building blocks for NAD^+^ and ADP-ribose [35]. We showed that dissociation of Fer from PARP-1 through downregulation of the kinase level with a selective siRNA, or through its direct dissociation from PARP-1 with its allosteric inhibitor (E260), which deforms the spatial structure of its KD, leads to deregulated activation of PARP-1 [25]. The deregulated activity of PARP-1 consumes cellular NAD^+^ and ATP, thereby contributing to cellular ATP depletion and energy crisis. The combined causes of ATP depletion, due to mitochondrial impaired activity on one hand, and deregulated nuclear activity of PARP-1 on the other hand, converge to evoke deregulated, ongoing autophagy which results in the onset of necrotic death in Fer/FerT targeted cancer cells [25]. Thus, the ability to simultaneously target through a selective Fer/FerT inhibitor, mitochondrial and nuclear metabolic processes may underlie the dramatic cytotoxic effect obtained when aggressive malignant cells such as metastatic and pancreatic ductal adenocarcinoma cells (PDAC) are being treated with the selective Fer/FerT inhibitor, E260 [25]. Following on from that said above, one could wonder from where the regulatory impact of Fer/FerT on PARP-1 stems. Here, again, the regulatory link between Fer/FerT and PARP-1 may originate from their regulatory roles in spermatogenic cells. The expression of FerT during spermatogenesis starts to rise in pachytene primary spermatocytes, which marks the prophase of the first meiotic division [20]. Events taking part at that stage include elaboration of the synaptonemal complex, assembly of the recombination nodules, homologous chromatid exchange, and resolution [36]. PARP-1 is most probably playing an important role, together with other DNA-repair regulators, in ensuring the maintenance of chromosomal DNA integrity following the active recombination stages. However, excessive activation of PARP has been shown to contribute to the pathogenesis of several diseases associated with oxidative stress, which has been known to play a fundamental role in the etiology of male infertility [37]. Hence, while the activity of PARP-1 might be important during meiotic homologous recombination, it has to be restrained and accurately modulated. The unique modulatory role of Fer/FerT on PARP-1 might be exploited by malignant cells for restraining their DNA damage repair machinery, thereby enabling the accumulation of an adverse DNA mutational level.

Collectively, the common metabolic ground laid by us between sperm and malignant cells opens new avenues for further exploring reprogrammed metabolic circuits in cancer cells, whose roots originate from unique metabolic pathways adopted by sperm cells in order to achieve and fulfill their very unique functional goals.

## 3. Roles of Fer in Modulating Sperm Functions

The acrosomal reaction (AR) in mammalian sperm is a regulated process which is essential for sperm penetration into the egg. It is widely accepted that physiological AR occurs as a result of the interaction between intact sperm and the egg zona pellucida (ZP), although it has been suggested that fertilizing mouse sperm can initiate their AR before contact with the ZP [38].

In order to interact with the ZP and undergo the AR, the mammalian sperm must first undergo a serial of biochemical processes in the female reproductive tract called capacitation (rev. in [39]). In previous studies, we showed that actin polymerization to produce F-actin occurs during sperm capacitation, and the F-actin is dispersed immediately prior to the AR [40]. The process of actin polymerization is mediated by phospholipase D (PLD) and Ca^2+^/calmodulin-dependent protein kinase II (CaMKII) [41]. PKA and tyrosine kinase are two key kinases involved in the capacitation process [42].

Sperm capacitation is accompanied by a rapid increase in bicarbonate influx and activation of soluble adenylyl cyclase to produce cAMP [43], leading to PKA activation and an indirect increase in protein tyrosine phosphorylation [42]. Since PKA is a serine/threonine kinase, it cannot directly execute this tyrosine phosphorylation process. Accordingly, it was suggested that Fer is responsible for the capacitation-associated protein tyrosine phosphorylation in murine sperm [44].

Spontaneous AR (sAR) can occur under some conditions, and sperm samples with a high proportion of sAR result in poor success in human IVF [45]. Several mechanisms protect sperm from sAR. It was shown that CaMKII prevents sAR in mouse sperm by interacting with MUPP1 [46]. However, its mechanism of action is not known. We also showed that inhibition of CaMKII in bovine sperm induces sAR [41]. The production of F-actin during capacitation protects the sperm from sAR. We showed that activation of the actin severing protein gelsolin induces a significant increase in sAR [47]. Moreover, inhibition of CaMKII or PLD during sperm capacitation caused inhibition of F-actin formation and an increase in sAR. Spermine, which leads to PLD activation, was able to reverse the effects of CaMKII inhibition on sAR increase and F-actin decrease [41]. Furthermore, the increase in sAR and the decrease in F-actin caused by the inactivation of the PLD pathway were reversed by activation of CaMKII using hydrogen peroxide or by inhibiting protein phosphatase 1, which enhances the phosphorylation and oxidation states of CaMKII [41]. In order to fully activate actin polymerization and prevent sAR, both forms of CaMKII, p-CaMKII and oxidized CaMKII, should be activated. These results indicate that two distinct pathways lead to F-actin formation in the sperm capacitation process which prevents the occurrence of sAR. Calcium channels mediate sAR in bovine sperm including the sperm-specific cation channel CatSper [48]. NMDA-type glutamate receptor mediates sAR in newt sperm [49]. The knockout of mice to β-Defensin [49], of the lipocalin family [49], or to the aldehyde dehydrogenase ALDH4A1, a key enzyme in mitochondrial prolin metabolism [50], shows an increase in sperm sAR. Surprisingly, in hamster sperm [51], and in equine sperm [44], bicarbonate increases sAR. It is known that bicarbonate activates soluble adenylyl cyclase to produce cAMP to activate PKA. In bovine sperm, PKA is indirectly involved in CaMKII activation leading to protein tyrosine phosphorylation and sperm capacitation [52] and preventing sAR [41]. We also showed that inhibition of PKA enhanced sAR in bovine sperm [53]. Src Family kinase (SFK) regulates PKA activity and sAR in chicken sperm [54]. In *C. Pyrrhogaster* sperm, activation of soluble adenylyl cyclase promotes sAR, whereas inhibition of PKA inhibits sAR [55]. In pig sperm, upregulation of the cytochrome C prevents sAR indicating that mitochondrial activity protects sperm from sAR [56]. This observation supports our notion regarding the regulation of the mitochondrial comp. I by Fer, in which its inhibition promotes sAR [26]. The precise regulation of the mitochondrial electron transport chain controls the production of ROS and protects the sperm from sAR. Thus, Fer, as an important regulator of mitochondrial activity, is responsible for providing ATP for various sperm functions leading to proper fertilization.

In our recent study, we show that inhibition of Fer inhibits actin polymerization during sperm capacitation resulting in the occurrence of sAR [26]. Actin polymerization during capacitation is essential for the development of hyperactivated motility [57] which is a known capacitation marker. Thus, we concluded that the regulatory, actin-polymerization-promoting activity of Fer supports proper sperm capacitation by protecting sperm from sAR. Interestingly, one of the ways through which Fer promotes actin polymerization in sperm is by restraining the activity of the phosphatase PP1, thereby maintaining the activation state of p-CAMKII [26]. We found that this ability of Fer to restrain the activity of PPI is exploited by cancer cells in their attempt to deregulate their cell cycle progression. Thus, suppression of the activity of PPI with Fer enables the deregulated activation of key cell cycle promoting factors in malignant cells. This was primarily manifested by the hyper-phosphorylation and inactivation of pRB, the tumor suppressor and cell cycle regulator [58].

## 4. Oxidative Stress and Male Infertility: Current Knowledge of Pathophysiology and the Role of Antioxidant Therapy in Disease Management

Oxidative stress is currently being considered as the main cause of male infertility (Rev. by [59]). Although low ROS is essential for the onset of sperm activating processes such as capacitation [60], its increased level disturbs sperm functions, thereby leading to male infertility by mechanisms such as lipid peroxidation and DNA damage [61]. The levels of ROS are therefore precisely regulated in sperm, mainly by superoxide dismutase, which coverts superoxide to H_2_O_2_ [62], and by catalase, which decomposes H_2_O_2_ [63]. Reactive oxygen species (ROS) is formed during sperm capacitation which is important for the activation of CaMKII [41] and PLD [53], two enzymes involved in actin polymerization during sperm capacitation. We show here that treatment of bovine sperm with 50µM H_2_O_2_ caused significant increase in CaMKII phosphorylation/activation, a state which is completely reversed by 100 µM H_2_O_2_ (Figure 1).

In cancer cells, Fer was shown to regulate ROS levels, and its downregulation causes a significant increase in ROS levels [23]. This activity of Fer suggests that Fer might function as a regulator of ROS levels, protecting sperm from damage, and this can also lead to sAR. This notion is based on our findings showing that the tyrosine kinase Src activates Fer [26], and ROS activates Src in human sperm [64]. In addition, as mentioned above, high concentration of ROS inhibits CaMKII (Figure 1) resulting in inhibition of F-actin formation and the occurrence of sAR [41].

In our recent study, we show that sAR induced in sperm by Fer inhibition can be restored by blocking Ca^2+^ channels including the sperm-specific Ca^2+^ channel CatSper [26]. These data suggest that CatSper is regulated by Fer which inhibits its activity. Thus, Fer could prevent the onset of sAR by simultaneously potentiating actin polymerization and F-actin formation on one hand, and by the restraining of ROS production and/or Ca^2+^ influx on the other hand. The increase in intracellular Ca^2+^ activates the F-actin severing protein gelsolin leading to F-actin dispersion and sAR [47]. There is contradiction in the literature regarding the regulation of CatSper by PKA. The group of Orta suggested that CatSper is regulated by PKA [65], whereas the group of Wang claim that CatSper is not activated by PKA [66]. We claim that CatSper is regulated/inhibited by Fer which is localized downstream to PKA (see model Figure 2), suggesting that PKA regulates CatSper indirectly via activation of Fer which inhibits CatSper.

Collectively, we show here that Fer and FerT govern several key regulatory pathways in sperm, thereby supporting proper sperm maturation and function. Some of these regulatory roles are harnessed by cancer cells to support their malignant transformation and oncogenicity (Figure 2).

## Figures and Tables

**Figure 1 ijms-24-05256-f001:**
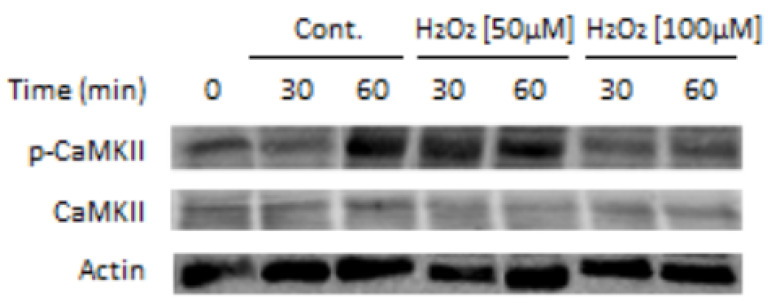
CaMKII phosphorylation/activation is affected by H_2_O_2_: Bovine sperm (1 × 108 cells/mL) were incubated under capacitation conditions (mTALP buffer) for 30 and 60 min in the absence (Cont.) or presence of H_2_O_2_ [50 µM, 100 µM]. Proteins were extracted and analyzed by Western blot using anti-phospho-CaMKII (Thr-286) (p-CaMKII), anti-CaMKII (CaMKII), and anti-actin (Actin) (loading control) antibodies. The results shown are representative of three independent experiments. See procedure information in [41].

**Figure 2 ijms-24-05256-f002:**
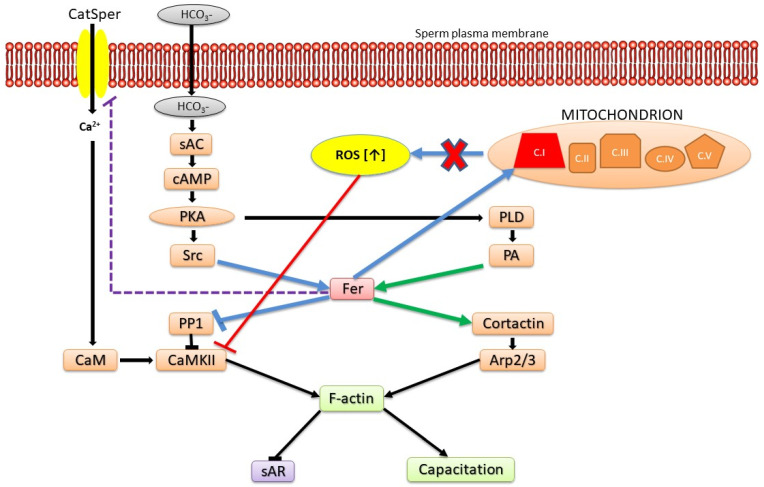
Model: Roles of Fer in modulating sperm functions. Prior to the initiation of sperm capacitation, PKA is activated by cAMP generated by the bicarbonate-activated soluble adenylyl cyclase (sAC). PKA phosphorylates/activates Src and PLD, two factors that mediate Fer activation. Activated Fer acting on several levels inhibits PP1 and regulates Ca^2+^ influx via CatSper leading to CaMKII activation and actin polymerization. Simultaneously, Fer also activates cortactin leading to Arp2/3 activation and F-actin formation. In addition, Fer regulates mitochondrial respiration via complex I, restrains ROS production, thereby prevents CaMKII inhibition by high levels of ROS. Blue arrows and interactions denote common pathways to sperm and malignant cells.

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
