# Peer review of "Fer and FerT: A New Regulatory Link between Sperm and Cancer Cells"

_ijms, 2023, doi:10.3390/ijms24065256_

Round 1

Reviewer 1 Report

Fer and its variant-FerT are non-receptor tyrosine kinases expressed specifically in sperm and cancer cells. The authors review similarities and differences in these two systems to learn about their modes of regulation and functional involvement. Fer and FerT, its truncated isoform, participate in actin-cytoskeleton integrity and function and modulate PARP-1 and the PP1 phosphatase. In addition, Fer and FerT appear to have regulatory functions linked to the metabolic state of sperm and cancer cells.

The review is interesting and helpful, as it provides novel perspectives of the regulation of complex responses in these two cell systems.

Even though this is a review, the authors mostly cite their own work without including pertinent contributions from other groups. An example is the controversies surrounding the regulation of CatSper by phosphorylation.

The authors should also mention other sAR preventing mechanisms related to the ionic permeability state of sperm and include the pertinent references. Before publishing their review, the authors should be more inclusive.

Author Response

More citations were added on sAR (see p.9 lines 9-20) and on the relationships between CatSper and Fer (see p.10 lines 30-31 and p.11 lines 1-4)

Reviewer 2 Report

The review presented here proposes the link between sperm and malignant cells having Fer and FerT as mediators.

In the model proposed in this review, the authors show that an increase in ROS inhibits CaMKII. The foregoing is based on an experiment (figure 1) where hydrogen peroxide affected the activity of CaMKII, as expressed by the authors. However, the result presented in this figure only shows the effect of hydrogen peroxide on CaMKII protein levels. No experiment allowed us to conclude about the activity of this protein. This must be corrected.

Author Response

The information on activated/phosphorylated CaMKII is already in Fig. 2. There is now a better explanation of this point (See Fig.1 legend).